# Granulomatous Cheilitis or Tuberculid?

**DOI:** 10.3390/antibiotics11040522

**Published:** 2022-04-14

**Authors:** Georgi Tomov, Parvan Voynov, Svitlana Bachurska

**Affiliations:** 1Department of Periodontology and Oral Mucosa Diseases, Faculty of Dental Medicine, Medical University of Plovdiv, 15-A “Vasil Aprilov” Blvd, 4002 Plovdiv, Bulgaria; 2Plastic and Reconstructive Surgery Division, UNI Hospital, 100 Georgi Benkovski Str., 4500 Panagyurishte, Bulgaria; ppvoynov@gmail.com; 3Department of Pathology, National Oncology Hospital, 6 Plovdivsko Pole Str., 1756 Sofia, Bulgaria; svitba@gmail.com

**Keywords:** granulomatous cheilitis, tuberculid, IGRA, TST, antibiotic treatment

## Abstract

The granulomatous cheilitis (GC) presents a heterogeneous group of disorders characterised by a granulomatous inflammation/reaction of the lips to various stimuli. Numerous etiologies have been proposed, including genetic, immunologic, allergic and infectious. Among the secondary causes of GC, an infection by Mycobacterium tuberculosis (MBT) should be considered. In such cases, the GC could be the clinical presentation of a tuberculid resulting from a hypersensitivity reaction to an underlying focus of active (ATBI) or latent tuberculosis infection (LTBI). This communication describes an immunocompetent patient diagnosed with GC resulting from tuberculid, who responded well to Isoniazid monotherapy.

## 1. Introduction

Many infectious pathogens may provoke a granulomatous response in the head and neck region and need to be considered a potential aetiology of granulomatous cheilitis (GC) [1]. Among all, Mycobacterium tuberculosis (MBT) is the most common infectious causative agent of granulomatous disease worldwide [2]. The active tuberculosis infection (ATBI) has a more significant burden of MBT than latent tuberculosis infection (LTBI) and acts as an infection reservoir. In contrast, the LTBI is “a state of persistent immune response to stimulation by MBT antigens with no evidence of clinically manifesting ATBI” [3]. An additional aspect of ATBI and LTBI is the so-named tuberculids which are lesions of the skin or mucous membrane resulting from type III or type IV hypersensitivity to mycobacterial antigens disseminated from active or latent tuberculosis foci [4]. MBT cannot be isolated from tuberculid lesions [4]. Tuberculids often develop in patients with asymptomatic tuberculosis infection, and few authors consider GC related to ATBI or LTBI as tuberculid due to both lack of detectable MBT in the lip lesion and positive response to antituberculosis treatment [5,6].

This communication aims to present an immunocompetent patient with granulomatous cheilitis diagnosed as tuberculid based on a positive interferon-gamma release assay (IGRA) test and positive intradermal tuberculin test (TST) and treated successfully with Isoniazid monotherapy.

## 2. Case Presentation

A 29-year-old female without underlying co-morbidities presented with indolent, slow-progressive upper lip swelling for five months. Other local or systemic symptoms did not accompany it. The primary diagnosis established by the dermatologist was allergic cheilitis (the patient was working in a cosmetic studio), but the swelling did not subside with antihistamines. The local examination revealed a significantly enlarged upper lip with erythematous colour and tenderness on palpation. The lip contour and the philtrum were deformed (Figure 1). The nasal mucosa, the floor of the mouth, throat, and tongue were normal. There was no facial nerve palsy or fissured tongue to support Melkersson–Rosenthal syndrome. A provisional diagnosis of Miescher granulomatous cheilitis was made.

Additional diagnostic tests were performed to exclude infectious or another origin of the lesion (HIV, HSV, Candida spp., Streptococcus spp.). Laboratory data on haemoglobin, blood cells count, liver and kidney function were within normal limits. The levels of the C1-inhibitor and C4 complement factors were normal and ruled out angioedema. The interferon-gamma release assay—IGRA (QuantiFERON-TB gold test) and the intradermal tuberculin test (TST) were positive, but the chest radiography was normal. Histopathological examination of the upper lip tissue biopsy revealed dilation of lymph vessels and a sparse perivascular lymphoplasmacytic infiltrate (Figure 2, red arrow) admixed with a few histiocytes, some of which tend to form small tuberculoid granulomas. (Figure 2, blue arrow). Both histochemical studies for MBT (Modified Acid-fast Stain) and fungi (PAS) were negative. The sputum was also found negative for the MBT. 

The constellation of histological and laboratory findings was sufficient to support a final diagnosis of tuberculid. The patient was admitted to a phthisiatry unit and was prescribed monotherapy of Isoniazid (300 mg daily) for six months. Before starting the treatment, the patient was tested for HIV, with a negative result. After two months, the lip swelling started slightly to reduce. After five months, the condition was nearly resolved (Figure 3) but the therapy continued until a negative IGRA test was obtained. One year after treatment, the results are stable and the patient is under regular monitoring by an oral pathologist and phthisiatrist.

## 3. Discussion

Although most of the studies excluded MBT as a potential cause of the granulomatous response of the lips [7], the relationship with tuberculosis has been described in a few publications. Kavala M. et al. reported one case of tuberculid presented as GC and improvement following anti-tuberculous chemotherapy for three months [5]. Bhattacharya M. et al. reported a child with GC, described as tuberculid secondary to asymptomatic tuberculosis [6]. Other authors described also angular cheilitis as a secondary manifestation of asymptomatic tuberculosis in adult patients in Cuba [8] and Morocco [9].

There is no established golden standard test for diagnosing tuberculids [4]. In contrast with the active tuberculous lesion, the lack of bacterial burden in tuberculids works against any diagnostic strategy based on the identification of the bacteria in loco. Even tests based on PCR of skin biopsy such as GeneXpert, whose accuracy is beyond any doubt, may demonstrate mycobacterial DNA in near 50% of the cases of tuberculids. [4] The diagnosis of tuberculids is rather indirect and relies on the evidence of cellular immune response to mycobacterial antigens [4,10]. The available diagnostic strategies are summarised in Table 1.

In all reported cases of tuberculids associated with asymptomatic tuberculous infection and manifested as GC, the histological evaluation revealed both granulomatous inflammation and lack of MBT [5,6,8,9]. The underlying tuberculous infection was proved based on microbiological confirmation of MBT in sputum and radiography findings. In our case, the histological findings met the criteria for granulomatous inflammation, but the sputum was negative for MBT and the chest radiography was found to be normal (no CT scan was done at that moment, and the authors consider this as a limit of the presented study) and the underlying focus was not found. The extensive investigation revealed no systemic symptoms and signs such as a cough, pain in the chest, coughing up blood or sputum, weakness or fatigue, weight loss, no appetite, chills, fever or sweating at night. LTBI was established as a diagnosis based on both positive IGRA and TST following the current recommendations. [11] Nevertheless, correct diagnosis of LTBI is challenging as normal chest radiographs are frequently reported, and different observers often dispute the interpretation of chest radiography results [12]. What is more, the negative mycobacterial culture results do not exclude the possibility of LTBI. Studies emphasised that the spectrum of human tuberculous infection lies within the continuous dynamics of bacterial metabolic activities and hosts’ immunological responses [13]. For these reasons, nowadays, we rely much more on molecular detection. The most commonly used tests for LTBI diagnosis nowadays are the intradermal tuberculin test and IGRA [11]. A critical moment, however, is that TST results could be affected by complex factors such as age, the nutritional and immunological status of the individual, the time interval between antigen exposure and the test performance, prior BCG vaccination, immunosuppression, genetic background, and cross-reactivity with environmental non-tuberculosis mycobacteria [11]. The IGRA is a whole blood assay developed approximately 18 years ago to detect the IFN-γ produced in vivo by sensitised T cells after in vitro stimulation with mycobacterial antigens. Currently, two tests are available—ELISpot-based T-SPOT.TB (Oxford Immunotec, Abingdon, UK) and the ELISA-based QuantiFERON Gold In-Tube (QFT-GIT; Qiagen, Hilden, Germany). The greatest advantage of IGRA over TST is that they are not confounded by prior vaccination with BCG. Whilst IGRA had major positive implications for the diagnosis of LTBI, it has limitations as well. Sensitivity, whilst higher than TST, is typically limited, particularly in key subgroups such as children and immunocompromised patients (HIV positive for example) [11]. In our case, the patient is immunocompetent and the correlation of positive IGRA test and positive TST, together with the histological interpretation of the lesion as tuberculide (a hypersensitivity phenomenon not associated with MBT presence), were found sufficient to support a diagnosis of LTBI, although the primary location of the infection was not found.

The successful treatment of the tuberculids is very much dependent on the possibility to treat the underlying tuberculous infection [4,10,14]. There is a consensus, however, that the tuberculids characteristically respond to appropriate anti-tuberculous treatment, even when no underlying focus of tuberculosis is found [15]. Since the tuberculids could appear both in individuals with active or latent tuberculosis, the patients should undergo a meticulous clinical and paraclinical evaluation to rule out ATBI before initiating drug therapy for LTBI, since monotherapy with Isoniazid, the most common frontline therapy for LTBI, would be highly inappropriate in the context of active tuberculosis [11]. In 2018, the World Health Organization issued updated guidelines on the treatment of LTBI in children and adults living in countries with high and low incidence rates of tuberculosis [3]. According to the guideline, to minimise the risk of acquired drug resistance, treatment for LTBI includes only one or two antibiotics [3]. This approach presupposes a low risk of acquired resistance based on the proof of limited replication among the insignificant number of viable bacteria in LTBI [14,16]. Additionally, the individual risk of reactivation must be balanced against the potential risks of developing treatment-related adverse events [11]. Therefore, patients with recently acquired LTBI should be evaluated for preexisting medical conditions that may increase the risk of such adverse events. In the presented case the medical history of the patient was within the norm and the HIV status was tested before starting the treatment with Isoniazid. However, few studies that evaluated the effect of Isoniazid monotherapy on the immune response of people with LTBI have revealed conflicting results [17,18]. Such conflicting data could be associated with variables such as the prevalence of TB, antibiotics used, treatment adherence, type of assay (QuantiFERON varieties vs. TB-SPOT), incubation periods (short vs. long), and antigen types (proteins vs. peptides).

The positive clinical response of the granulomatous cheilitis to the Isoniazid monotherapy fulfilled additional diagnostic criteria for tuberculid associated with LTBI but the lack of specific recommendations regarding treatment monitoring makes the results rather empirical. The need to assess the effectiveness of LTBI therapy has led to the search for immunological markers for this purpose [19]. Since the diagnosis of LTBI is established on the basis of the immune response of individuals to MBT antigens, the immunological tests used for the diagnosis have also been adopted to assess response to drug treatment [20]. Based on its sensitivity and specificity, IGRA has also been suggested for use as a treatment monitoring tool [21,22]. Several studies have shown that patients who are IGRA negative on completion of anti-tuberculous therapy experienced complete clinical and microbiological recovery [23,24]. Another cohort study showed that patients who were IGRA-positive at the end of treatment developed tuberculous reactivation, while those who were IGRA-negative did not develop tuberculous reactivation for 2 years of follow-up [25]. In our clinical case, the effectiveness of LTBI therapy was based both on the clinical evaluation of the lip lesion and on a negative IGRA test in the sixth month of treatment but the case monitoring was prolonged to one year. 

## 4. Conclusions

The granulomatous cheilitis interpreted in this case report as a tuberculid represents one of the occult faces of the LTBI. The diagnosis of tuberculid was not self-evident at the very beginning of the diagnostic process and a significant number of possible differential diagnoses were ruled out—Melkersson–Rosenthal syndrome, Miescher’s granulomatous cheilitis, sarcoidosis, angioedema, infections of another origin, etc. Often beginning as a complex-mediated immune reaction in distant sites, tuberculids evolve into a granulomatous inflammatory response that is MBT-free and very similar to other granulomatous lesions. Misdiagnosis (GC of another origin) combined with improper therapy (corticosteroids, antihistamines, anti-inflammatory drugs) will blur or even deteriorate the clinic and the prognosis. The final diagnosis of that particular case was based on positive IGRA and TST and confirmed by further improvement with anti-tuberculous therapy. Considering namely the rarity of this condition (only a few reported cases worldwide), we want to emphasise that this potential correlation should be taken into consideration in all cases of granulomatous cheilitis. The paper also raises the question of the lack of established diagnostic and therapeutic guidelines regarding this unusual manifestation of tuberculous infection.

## Figures and Tables

**Figure 1 antibiotics-11-00522-f001:**
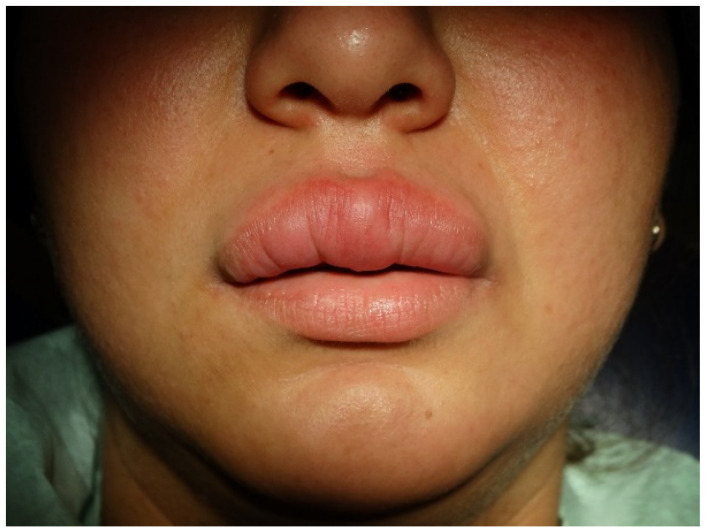
A pretreatment view of the upper lip shows diffuse erythematous swelling tender on palpation. The lip contour is blurred.

**Figure 2 antibiotics-11-00522-f002:**
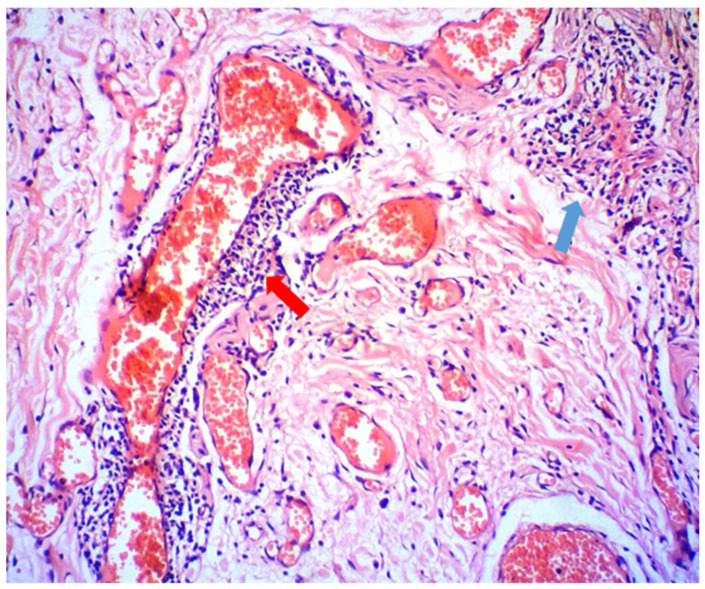
Perivascular inflammatory infiltration of lymphocytes (indicated with a red arrow) and small tuberculoid granuloma (indicated with a blue arrow) (HE, 100×).

**Figure 3 antibiotics-11-00522-f003:**
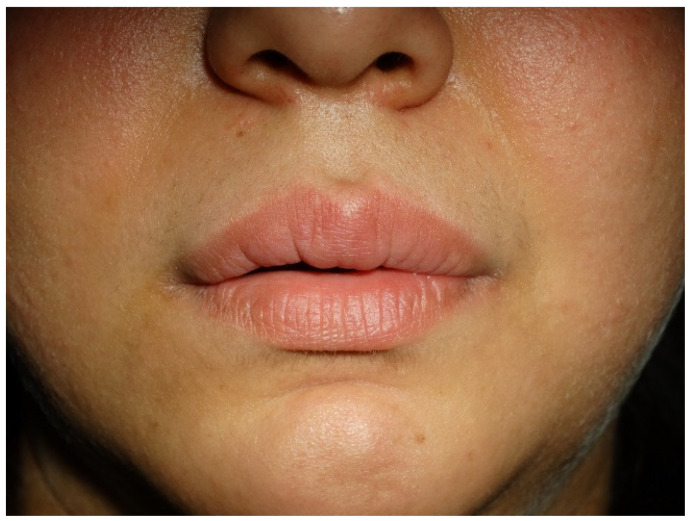
Posttreatment view of the patient with marked improvement of upper lip swelling and erythema after five months of Isoniazid monotherapy.

**Table 1 antibiotics-11-00522-t001:** Diagnostic strategies for tuberculids.

Diagnostic Test	Expected Results
Biopsy for histological evaluation	Granulomatous inflammation ^1^ (+)
Biopsy, including staining for acid-fast bacilli	Usually negative (+/−)
Tissue culture for mycobacteria	Negative (−)
PCR of skin biopsy	May demonstrate mycobacterial DNA in near 50% of the cases (+/−)
Intradermal tuberculin test (TST)	Positive (+)
Interferon-gamma release assay (IGRA)	Positive (+)
Extensive investigation for underlying tuberculosis, in addition to family history and clinical features	Tuberculids characteristically respond to anti-tuberculous treatment, even when no underlying focus of tuberculosis is found (+)

^1^*Granulomatous—a histological term for a chronic inflammatory pattern characterised by localised aggregations of histiocytes with or without other inflammatory cells, with or without necrosis, with or without vasculitis, with or without calcification, with or without foreign bodies. Granulomas may be due to infection, chronic inflammatory disease, or foreign body reaction*.

## Data Availability

Not applicable.

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
