# Peer review of "Granulomatous Cheilitis or Tuberculid?"

_antibiotics, 2022, doi:10.3390/antibiotics11040522_

Round 1
Reviewer 1 Report
There is one major issue in this report, in relation with the following statement made by the author: "The clinical, histological and laboratory findings constellation was sufficient to support a final diagnosis of granulomatous cheilitis related to LTBI".
The co-occurence of LTBI, which is quite frequent in the general population (at least 30-40% depending on the region you live), with granulomatous cheilitis, does not imply causality.
Next, by definition, LTBI is not an active disease, that means there can not be any clinical signs during LTBI (except perhaps for primo-infection?the authors could have discussed that). I agree that this is a poor definition which is surely going to evolve (remote versus recent non active infection), but recommandations are edicted this way.
Secondly, IGRA is not approved at all to make a diagnosis of active tuberculosis, which could be suspected on the basis of the histopathology results. It is only a marker of past contact with M.Tub bacilli. Unfortunately, no mycobacteriological examinations or MTBRif assay of the lip samples have been made in the present case, so the authors cannot really conclude to the absence/presence of mycobacterium tuberculosis.
Third, chest X Ray is really a poor exam to definitely rule out Tuberculosis, and we would have appreciated to get chest CT scan results.
Fourthly, IGRA follow-up has been evaluated in many prospective studies. It is definitely not a good biomarker to assess any response to antituberculous drugs.
Finally, even nowadays, histopathological data may be sometimes the only evidence for active tuberculosis in some cases, when no bacilli are highlighted in numerous samples (FAST, culture growth, XpertMTBRif, etc...). In that case, a drug regimen with HRZE at the begining remains the standard (depending on sensitivity)
The authors should really consider some differential diagnosis or perform other examinations to confirm active tuberculosis.
Author Response
Dear Madam/Sir,
We appreciate your efforts and precious time in reviewing our paper and providing valuable comments. All authors have carefully considered the remarks and tried their best to address every one of them.
Please see the attachment with our point-by-point responses (highlighted in red).
Kind regards

Reviewer 2 Report
This is a case report from Tomov et al which describes a 28-year old female immunocompetent patient diagnosed with granulomatous cheilitis, a form of latent tuberculosis infection. The authors showed that an Isoniazid monotherapy with 300 mg daily over a six-month period was efficient to treat the pathology which is confirmed by a negative interferon-gamma release assay (IGRA) at the end of the treatment. Further, a year after the treatment, the results were stable. This is an important work which could help other clinicians to diagnose latent tuberculosis infection through observing an upper lip swelling. Further, the work also highlights an efficacity of a monotherapy approach in a patient without underlying co-morbidities, an information which will be helpful to disseminate to other clinicians around the world.
Minor comments
Figure 2 and 3 have to be adjusted
Line 90: correct the sentence: In Cuba and Morocco respectively
Author Response
Dear Madam/Sir,
We appreciate your efforts and precious time in reviewing our paper and providing valuable comments.
Please see the attachment with the revised file (corrections are highlighted in red).
Kind regards

Reviewer 3 Report
In the current manuscript, Tomov et al. correlated a case study between granulomatous cheilitis and latent tuberculosis (LTBI). The authors showed the data of a 28-year-old female patient with progressive upper lip swelling. The patient didn't have any underlying co-morbidities, which are often associated with tuberculosis, including HIV. Overall, the correlation made by the authors is reasonable. However, due to the very limited nature of the data provided, including the lack of IGRA and TST results, the manuscript is not suitable for publication in MDPI-Antibiotics.
Major Comments:
- The patient was prescribed Isoniazid. However, according to CDC, the most standard regimen for LTBI includes both, Isoniazid and Rifampicin (https://www.cdc.gov/tb/topic/treatment/ltbi.htm). The authors fail to mention or discuss the rationale for using Isoniazid therapy.
- Isoniazid therapy resolved the clinical implications of the patient's lip abnormality. However, due to the lack of control experiments/clinical tests post-treatment, it is difficult to make a strong correlation between lip abnormality and LTBI.
- To make any clinical correlation between two pathologies, in this case, GC and LTBI, it is important to have a case series rather than a case report. Importantly, the current case report has a single piece of evidence (which is missing the key result), making the study's conclusions neither scientifically reasonable nor justified.
Minor Comments:
Line-33: Rewrite the sentence. There are no distant sites of latent TB.
Line-34: Unclear sentence, rewrite.
Line-35: Replace the word 'silent.'
Line-60: It is unclear as to what tissue was examined. Please state the result clearly.
Line-63: Explain the result better and separate Figure-2 from Figure-3. It is confusing when the single panel figures are pasted together, and the legends are given at the bottom. This can be easily made into one figure with clearly labeled panels.
Line-63: Figure-3, use arrows to show the discussed result.
Line-63: Expand PAS.
Line-89: The word 'respectively' is unnecessary.
Line-96: What is GH? or do the authors mean GC?
Line-133: Provide a reference for the statement.
Line 183: What is AGRA? or do the authors mean IGRA?
Author Response

(The authors gave the same response as above.)

Round 2
Reviewer 1 Report
The authors improved their case report, unfortunately the lack of igra antériority, the absence of mycobacteria culture or genxpert on the skin biopsy, and not having performed chest CT scan , are still major issues nowadays.
All the conclusions rely on old observations from times where genxpert or chest CT scan were not routinely performed. Current reports need those kind of examinations in order to improve our understanding of tuberculosis.
Finally, LTBI defining itself by an absence of clinical sign, I think the assumption of the authors is still erroneous.
Author Response
Dear Madam/Sir,
We appreciate your efforts and precious time in reviewing our paper and providing additional valuable comments. All authors have carefully considered the remarks and tried their best to address every one of them.
Please see the attachment with our point-by-point responses (highlighted in red) and the revised paper.
Kind regards

Reviewer 3 Report
The authors have addressed the major critics/comments on the papers. However, it is not clear if they have addressed all the minor comments. I tried to match my comments with the yellow highlighted changes in the manuscript but I could find all the changes. They should have responded the response to minor comments as they did for the major comments.
If all the minor comments were addressed, I think the authors have addressed all my concerns.
Minor Comments:
Line-33: Rewrite the sentence. There are no distant sites of latent TB.
Line-34: Unclear sentence, rewrite.
Line-35: Replace the word 'silent.'
Line-60: It is unclear as to what tissue was examined. Please state the result clearly.
Line-63: Explain the result better and separate Figure-2 from Figure-3. It is confusing when the single panel figures are pasted together, and the legends are given at the bottom. This can be easily made into one figure with clearly labeled panels.
Line-63: Figure-3, use arrows to show the discussed result.
Line-63: Expand PAS.
Line-89: The word 'respectively' is unnecessary.
Line-96: What is GH? or do the authors mean GC?
Line-133: Provide a reference for the statement.
Line 183: What is AGRA? or do the authors mean IGRA?
Author Response

(The authors gave the same response as above.)
